# Style-conditional Prompt Token Learning for Generalizable Face Anti-spoofing

## ABSTRACT

Face anti-spoofing (FAS) based on domain generalization (DG) has attracted increasing attention from researchers. The reason for the poor generalization is that the model is overfitted to salient liveness-irrelated signals. However, the previous methods alleviate the overfitting by mapping the images from multiple domains into a common feature space or promoting the separation of image features from domain-specific features and task-related features. This direct manipulation of image features inevitably destroys the semantic structure. If the text features of vision-language pre-trained (VLP) models (e.g., CLIP) are used to dynamically adjust the image features to gain a better generalization, we can not only explore a wider feature space but also avoid the potential degradation of semantic information. Specifically, we propose a FAS method of Style-Conditional Prompt Token Learning (S-CPTL), which aims to generate generalized text features by training the introduced prompt tokens to carry visual styles and use them as weights for classifiers to improve the model's generalization. Compared to the inherently static prompt token, we propose the dynamic prompt token, which can adaptively capture live-irrelevant signals from the instance-specific styles and increase their diversity through mixed feature statistics to further reduce the overfitting of the model. Thorough experimental analysis demonstrates that S-CPTL exceeds current top-performing methods in four distinct cross-dataset benchmarks.

## CCS CONCEPTS

• **Computing methodologies → Computer Vision**.

## KEYWORDS

Face Anti-Spoofing, Domain Generalization,CLIP, Prompt Learning, Style Condition

## 1 INTRODUCTION

Face Anti-Spoofing (FAS) plays a crucial role in safeguarding face recognition systems from presentation attacks (e.g., print attacks, digital replay and 3D synthetics masks etc.). Previous FAS [10, 12, 21, 36, 38, 39, 41, 43] have shown effectiveness in intra-domain scenarios, but may suffer from dramatic degradation when adapting to unseen domains.

**Unpublished working draft. Not for distribution.**

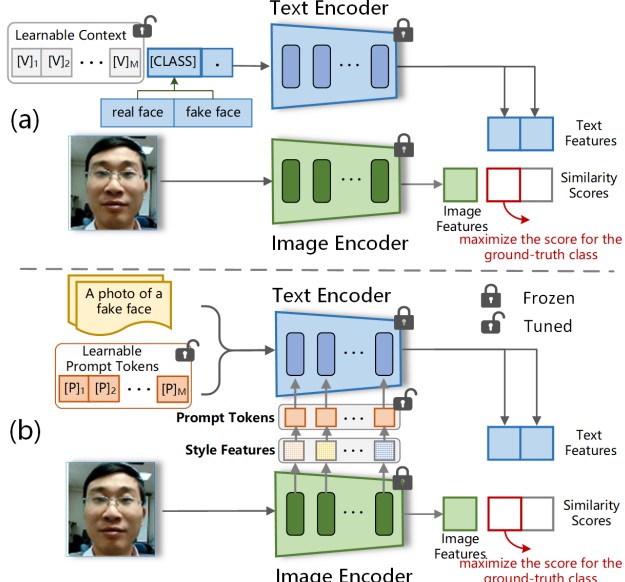

**Figure 1: (a) Existing method (i.e.CoOp [49, 50]) utilize prompt learning strategies to refine CLIP representations as prompts are exclusively developed through non-semantic vectors. (b) S-CTPL introduces style conditional prompts token that makes full use of the multi-level aligned vision-language knowledge to assist FAS tasks and achieve stronger generalization capabilities.**

To improve the generalization, the basic idea of existing methods always focus on learning domain-invariant representation from face images. Some methods [2, 8, 13, 27] aims to learn a generalized feature space by aligning the distributions among multiple source domains. The other methods [45, 53] utilize disentangled representation learning to isolate the liveness-related features from image features. However, such direct manipulation only on images will lead to insufficient semantic information and the destruction of semantic structure during the training.

We note that the core issue affecting the model's domain generalization is the classifiers are unable to effectively eliminate perturbation from liveness-irrelevant signals.Inspired by vision-language pretraining (VLP) (such as CLIP [26]), we propose to use text representations as weights for visual classifiers to enhance the generalization of FAS systems, which brings two benefits: 1) preventing the absence of semantic information and disruption of semantic structure, 2) enabling the learning of more refined visual representations and exploration of a broader feature space. Although CLIP demonstrates significant generalization capabilities, its extensive scale makes it inadvisable to engage in full model fine-tuning

for downstream task adaptation. This is attributed to the possibility that such fine-tuning could lead to the erosion of valuable knowledge obtained during the extensive large-scale pretraining phase and may introduce the risk of overfitting. Currently, the common methods for CLIP adapting to downstream tasks are CoOp [**?**] and CoCoOp [**?**], which are achieved by prompt learning, as shown in Figure 1. Compared to effectiveness in discriminating between fundamental categories such as "cat" and "dog" in image classification tasks that have explicit category semantics, FAS has no semantic meaning in the determination of whether a face is 'real' or 'fake', which restricts the direct use of prompt learning to obtaining representations with context categories. Due to liveness-irrelevant signals leading to poor generalization being entangled in image features, methods based on prompting learning(e.g. Co-CoOp), which adjust prompts based on complete visual features, fail to precisely support the classifier in avoiding interference derived from liveness-irrelevant signals.

Instead of prompt learning, we adapt CLIP to the FAS task via fixed prompt templates combined with prompt token learning, which makes full use of the aligned vision-language knowledge to assist FAS tasks and achieve stronger representation capabilities. Furthermore, since liveness-irrelevant cues are lying within style features, the key to FAS domain generalization is to prevent the models from overfitting to instance-specific styles. Furthermore, since liveness-irrelevant cues are lying within style features, the key to FAS domain generalization is to prevent the models from overfitting to instance-specific styles. In this paper, we provide the corresponding solution S-CPTL(*Style-conditional Prompt Token Learning*) for the generalizable representation learning problem: 1)Constructing prompt token learning framework for FAS tasks. Reducing the semantic gap in the model by transferring generalized language guidance from large-scale models as classification weights for visual features. **1) Constructing prompt token learning framework for FAS tasks.** Reducing the semantic gap in the model by transferring generalized language guidance from large-scale models as classification weights for visual features. **2) Generating instance-specific style condition.** We extract learnable style information, which can benefit the model by adaptively capturing live-irrelevant signals. Compared to the inherently static methods, S-CPTL can dynamic mix feature statistics with text representations to further reduce the overfitting of the model. The main contributions presented in this research are detailed as follows:

- We propose a novel style-conditional prompt token learning approach to achieve generalized face antispoofing by employing visual style as a learnable prompt token condition.
- we propose the instance-aware dynamic style condition generation module, which can adaptively capture live-irrelevant signals from the instance-specific styles, thereby reduce the overfitting of the model.
- We validate the proposed method under various settings, including zero-shot cross-domain generalization and unseen attack detection. Experimental results show that S-CPTL consistently achieves stronger effectiveness than competing methods.

## 2 RELATED WORK

### 2.1 FAS Methods on Intra-dataset

The essence of FAS is a defensive measure for face recognition systems and has been studied for over a decade. Some CNN-based methods [7, 22–24, 31] design a unified framework of feature extraction and classification in an end-to-end manner. Intuitively, the live faces in any scene have consistent face-like geometry. Inspired by this, some works [21, 28, 38, 44] leverage the physical-based depth information instead of binary classification loss as supervision, which are more faithful attack clues in any domain. With the popularity of high-quality 2d attacks, i.e., OULU-NPU [1], SiW [21], CelebA-Spoof [47] and high-fidelity mask attacks, i.e., MARsV2 [16], WMCA [6, 25], and HiFiMask [15**?** ] with more realistic in terms of color, texture, and geometry structure, it is very challenging to mine spoofing traces from the visible spectrum alone. Methods based on multimodal fusion [5, 6] have proven to be effective in alleviating the above problems. The motivation for these methods is that indistinguishable fake faces may exhibit quite different properties under the other spectrum. In order to alleviate the limitation of consistency between testing and training modalities, flexible modality based methods [40] aims to improve the performance on any single modality by leveraging available multimodal data. However, above methods are not specially designed to solve the domain generalization.

### 2.2 Domain Generalization for FAS

To address this issue, Domain Adaptation (DA) [19, 20, 32] aims to minimize the distribution discrepancy between the source and target domain by leveraging the unlabeled target data. However, the target data is difficult to collect, or even unknown during training. Domain Generalization (DG) can conquer this by taking the advantage of multiple source domains without seeing any target data. Currently, there is a consensus that minimizing the discrepancy of feature distributions between the source and target domains is a core factor in generalization. As such, most works based on adversarial learning [8, 27] or meta-learning [17, 18, 37] focus on learning domain-invariant representation from face images. Methods [30, 33, 45, 53] utilize disentangled representation learning to exploit the domain-specific features. However, most of these methods require domain labels, with the attendant efforts of label annotation. The domain labels of FAS tasks lacks semantics, while the diversity of domain criteria simultaneously blurs the boundaries of the domain. IADG [52] is the proposed as an alternative without need for domian labels, it improve generalization by reducing the sensitivity of instance-specific style features. In summary, existing methods mentioned above are typically developed in the situation that models are only guided by image data and corresponding image labels during training. Such an operation imposes the limitation of insufficient representative ability.

In the FAS scenario, FLIP [29] as an end-to-end finetune CLIP framework with an ensemble of prompt templates on FAS tasks was proposed. Nevertheless, we note that the prompt templates need to be manually adjust while the set of prompt template will cause large-scale pretraining phase. In addition, such fine-tune strategy will overlook the useful knowledge acquired in the pretraining phase and will potentially result in overfitting. In contrast, we prefer to

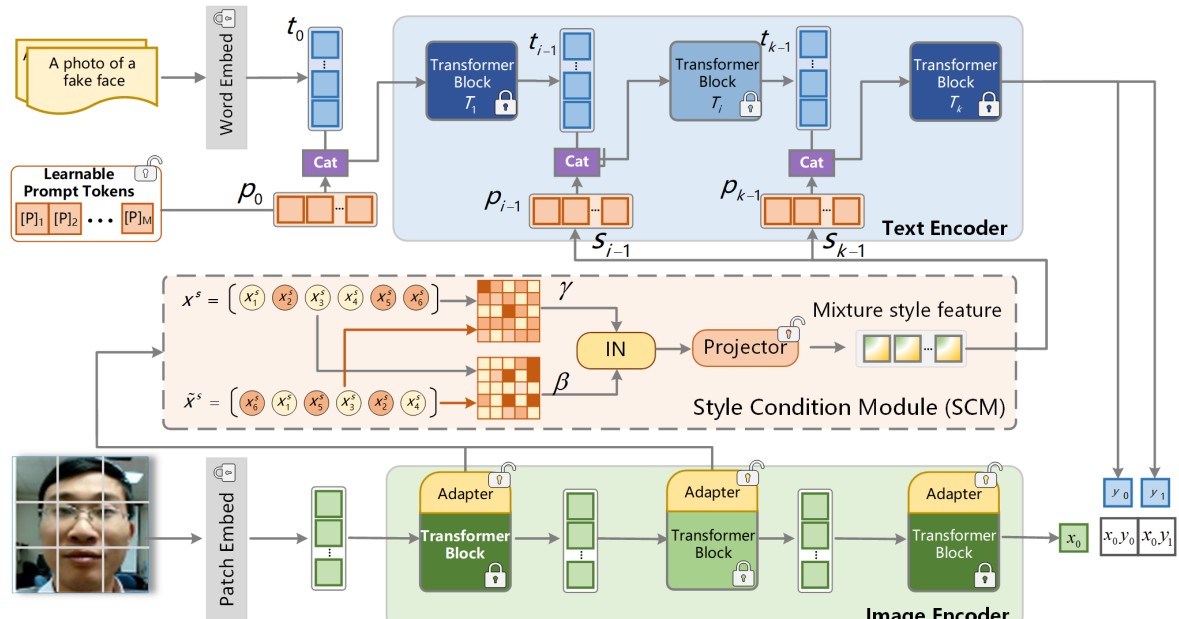

**Figure 2: Overall architecture of our proposed Style-Conditional Prompt Token Learning (S-CTPL) framework for DG FAS. Our S-CTPL is built on CLIP and realizes adaption to FAS tasks by leveraging prompt token learning with two main contributions: (1) Prompt Token Learning. S-CTPL generate generalized text features by training the introduced prompt tokens to carry visual styles and use them as weights for classifiers to improve the model's generalization.(2) Style Condition Module (SCM). SCM, which consists of bypass convolutional adapter and hybrid style augmentation module (HSAM), can dynamic mix feature statistics with text representations to further reduce the overfitting of the model.**

efficiently adapt from general images to detect more intricate face spoofing. We introduce a simple yet pioneering approach to prompt token learning to establish robust and extensive baselines for FAS.

## 2.3 Prompt Learning in VLP models

With increasingly strong application demands for visual language pre-trained(VLP) model, how to use a small number of learnable parameters to adapt large models to downstream tasks has been an important topic. Motivated by prompt learning in NLP, many studies have suggested adapting V-L models by training prompt tokens in an end-to-end approach. CoOp [50] and CoCoOp [49] are currently the common methods for CLIP adapt to downstream tasks. CoOp optimizes a continuous set of prompt vectors at its language branch to fine-tune CLIP for fewshot transfer. However, CoCoOp reveals that CoOp underperforms on novel classes. This issue is solved by conditioning prompts on image instances to ensure better generalization. MaPLe [11] is designed to enhance the alignment between vision and language representations through its implementation in both branches.

In summary, previous studies have established their efficacy in distinguishing basic classifications in image classification tasks that have explicit category semantics. However, the semantic meaning of FAS is not relevant in discerning whether a face is real or fake. In the FAS, blindly pushing FAS tasks towards generic large-scale VLP models will lead to suboptimal transfer outcomes.

## 3 S-CPTL: STYLE-CONDITIONAL PROMPT TOKEN LEARNING

### 3.1 Review of CLIP

CLIP [26] has revolutionized the field of visual representation learning, showcasing remarkable capabilities in capturing the intricate relationship between image and corresponding textual description. This model is uniquely characterized by using an image encoder and a text encoder to predict the correct image-text sample pairs. Formally, we define $\mathbb{D} = \{(x_i, t_i)\}_{i=1}^{B}$ as the collection of image-text sample pairs with $C$ categories, and $B$ is the number of images in a mini-batch . $x_i \in \mathbb{R}^{H \times W \times 3}$ is the image and $t_i$ indicate the corresponding text description.

**Encoding Image:** Image encoder initially divides the image into $N_v$ fixed-size patches which are projected to create patch embeddings $E_i^p \in \mathbb{R}^{N_v \times d}$, where $d$ is the hidden dimension of CLIP. Each patch embedding, along with a learnable token [CLS], passes sequentially through a Transformer or ResNet to obtain image representation $v_i \in \mathbb{R}^{d_v}$, where $d_v$ is the visual feature dimension,such as 768 for ViT-B/16 and 2048 for ResNet50. To obtain the final image representation, the class token of last layer is projected to a common V-L latent embedding space via a image projector.

**Encoding Text:** In CLIP, prompt tuning adapts the model to downstream tasks by using manually crafted templates to form the prompt while keeping the pre-trained image and text encoders fixed.

Given the dataset with category names $\{[CLASS]_c\}_{c=1}^C$, each class-wise text description is denoted as $t_c^{clip} = \{A\ photo\ of\ a\ [CLASS]_c\}$. Text encoder first tokenizes each word of the description $t_c^{clip}$ by assigning a specific numeric ID. Each sequence of tokens, which is enclosed within [SOS] and [EOS] tokens and limited a fixed length of 77, is further project to the word embedding $E_c^w \in \mathbb{R}^{77 \times d_l}$ and then passed on to a Transformer to generate textual representation $l_c^{clip} \in \mathbb{R}^{d_l}$, where $d_l$ is the textual feature dimension, such as 512 for text Transformer. Similar as encoding image, The final text representation is obtained by projecting the text embeddings corresponding to the last token of the last transformer block er to a common V-L latent embedding space via a text projector.

**Zero-shot Classification:** Based on the visual feature $v_i$ and textual feature $l_c^{clip}$, the prediction probability is calculated as:

$$p(y = c \mid v_i) = \frac{\exp\left(\text{sim}\left(v_i, l_y^{clip}\right)/\tau\right)}{\sum_{c=1}^C \exp\left(\text{sim}\left(v_i, l_c^{clip}\right)/\tau\right)} \quad (1)$$

where the sim($\cdot$) refers specifically to cosine similarity and $\tau$ is a temperature parameter. Our method involves the direct utilization of a pre-trained CLIP model.

## 3.2 Style-conditional prompt token learning

*3.2.1 Overview.* To efficiently steer a VLP to tackle downstream FAS tasks, we explore the novel prompt token learning framework. We reason that prior works [49, 50] that endeavor emphasizing prompt learning approaches are less suitable as the classification weights are dependent on a pre-defined category name set, leading to subpar performance when dealing with non-semantic category names (e.g., real/fake face). Furthermore, FAS is a much more refined vision task, the model needs to learn to characterize each instance rather than to serve only for some specific category. Therefore, we note that employing prompt tokens in the deeper transformer layers can systematically capture more stage-wise feature representations.

Fig. 2 presents an overview architecture of our proposed **S**tyle-**C**onditional **P**rompt **T**oken **L**earning (**S-CTPL**) framework. Our S-CTPL is built on CLIP consists of frozen image encoder $\mathcal{V}(\cdot)$ and text encoder $\mathcal{T}(\cdot)$ with $K$ transformer layers $\{\mathcal{V}_i\}_{i=1}^K, \{\mathcal{T}_i\}_{i=1}^K$, and adapts to FAS tasks via the learnable prompt token and a style condition module (**SCM**) to weaken the features' sensitivity to instance-specific styles. In the textual branch, we introduce learnable tokens $\{P_i\}_{i=1}^J$ in first $J$ transformer layers. When $J = 1$ the learnable token $P_1$ concat with fixed input token as the input to first transformer layer, and then new learnable tokens $\{P_i\}_{i=2}^J$ are further introduced in each subsequent transformer layers $\{\mathcal{T}_i\}_{i=1}^J$ of the text encoder up to a specific depth $J$. To adaptively capture live-irrelevant signals from the instance-specific styles, we propose SCM to dynamic mix feature statistics with text representations to further reduce the overfitting of the model. Concretely, SCM consists of bypass convolutional adapter and hybrid style augmentation module (HSAM). Firstly, in the visual branch, we use bypass convolutional adapter, which is placed parallel to the Multi-Headed Self-Attention and MLP blocks. Some of the features output from the Adapter continue to be passed on to the next layer

of transformers, while others are fed into HSAM for diversity enhancement.Below, we will describe the design of prompt token learning and the process of style condition module.

*3.2.2 Prompt Token Learning.* To obtain hierarchical contextual representation, we introduce learnable prompt tokens in the first $J$ (where $J < K$) layers of language branches. Through word tokenization and subsequent projection into word embeddings, the CLIP text encoder generates feature representations $W_0$ for textual descriptions, $W_0 \in \mathbb{R}^{d_l}$. During each stage, $W_i$ as the input for the subsequent $(i + 1)$ transformer block in the text encoder. In the meantime, $M$ learnable tokens $\left\{P^i \in \mathbb{R}^{d_t}\right\}_{i=1}^M$ is concatenated with $W_i$ to learn the context prompts in the language branch of CLIP. The input embeddings now follow the $\left[P^1, P^2, \cdots, P^m, W_0\right]$, where $W_0$ corresponds to fixed input tokens. New learnable tokens are further introduced in each transformer block $\mathcal{T}_i$ of the text encoder, the stage will be defined as:

$$[\_, W_j] = \mathcal{T}_i\left(\left[P_{j-1}, W_{j-1}\right]\right) \quad j = 1, 2, \cdots, J \quad (2)$$

The concatenation operation is denoted by $[\cdot, \cdot]$. After the $J - th$ transformer layer, the following layers process prompts from the previous layer:

$$[P_j, W_j] = \mathcal{T}_j\left(\left[P_{j-1}, W_{j-1}\right]\right) \quad j = J + 1, \cdots, K \quad (3)$$

When $J = 1$ the learnable token $P_i$ concat with fixed input token transformed into text features by text encoder. In the case of $J = 1$, the application of learnable prompt tokens are restricted to the input of the first transformer layer, which is same as CoOp.

*3.2.3 Style Condition Module.* With liveness-irrelevant cues present in style, the primary focus for FAS domain generalization is to curb model overfitting to instance-specific styles. The potential method not only can convert original image embeddings into a new space specifically tailored for FAS, but also reduce the contribution of liveness-irrelevant representations and pay more attention to task-related features during the V-L space alignment process. We introduce an adapter(i.e. Convpass [9]]) to style condition generation module, which can capture discriminative local information that are sensitive to domian shift. The adapter reconstructs the spatial structure of the token sequence and performs convolution on image patch embed token and class embed token individually. Benefits from the inherent locality of convolutional layers, the adapter can help to capture fine-grain visual information.

While adapter adapt base model to FAS task, it remains susceptible to domain shifts. In term of domain shifts is hidden in style variances, extracting the style feature and weaking it's influence can improve the domain generalization. Therefore, we refine style features through the normalization of feature tensors using instance-specific mean and standard deviation. Our method draws inspiration from Mixstyle [51], which suggest that mixing styles among training instances leads to the implicit synthesis of mixed novel style. The improved diversity in styles contributes to increased resilience against variations specific to individual domains, which inspires us to introduce style extraction.

However, the Mixstyle is implemented in the end-to-end finetune fundamental image classification, whose scenario is vastly different from ours in two ways. Firstly, Mixstyle reveals the relation between

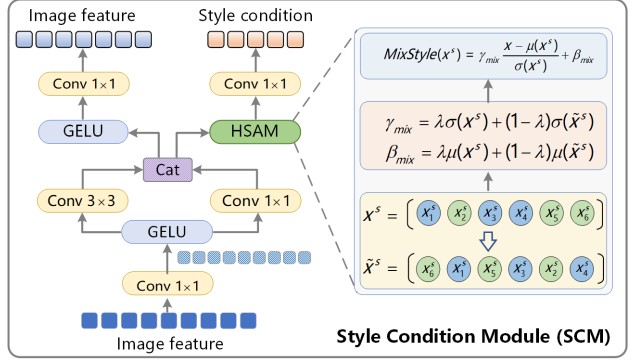

**Figure 3: The proposed Style Condition Module (SCM). SCM consists of bypass convolutional adapter and hybrid style augmentation module (HSAM)**

visual domian and image style, while we need to construct the link between visual domian, prompt and image style to fit into the CLIP fashion . Then, instead of application in general classification tasks, adapting CLIP for FAS need to new design to capture discriminative local information. The style condition generation module is placed side by side with the last convolutional layer of adapter. From the convolution with the same input and output channel, we get the image feature representation $X^* \in \mathbb{R}^{H \times W \times C}$, where H, W, C denote the height, width, and number of channels respectively. Then, we permute dimensions to obtain $\widetilde{X}^S \in \mathbb{R}^{C \times H \times W}$. Style extractor $S$ executes a type of style normalization through the normalization of feature statistics, which can be expressed as:

$$S(X^S) = \gamma \left( \frac{X^S - \mu_c(X^S)}{\sigma_c(X^S)} \right) + \beta \qquad (4)$$

$\mu_c(.)$ and $\sigma_c(.)$ are mean and standard deviation computed across the dimension within each channel of each image instance:

$$\mu_c(X^S) = \frac{1}{HW} \sum_{h=1}^{H} \sum_{w=1}^{W} \left( X^S_{chw} \right) \qquad (5)$$

$$\sigma_c(X^S) = \sqrt{\frac{1}{HW} \sum_{h=1}^{H} \sum_{w=1}^{W} \left( X^S_{chw} - \mu_c(X^S) \right)^2 + \epsilon}. \qquad (6)$$

In cases where domain labels are unknown, $\widetilde{X}^S$ is randomly sampled from the training data, and is simply obtained by shuffle operation. We computes the mixed feature statistics by

$$\gamma = \lambda \sigma(X^S) + (1 - \lambda)\sigma(\widetilde{X}^S) \qquad (7)$$

$$\beta = \lambda \mu(X^S) + (1 - \lambda)\mu(\widetilde{X}^S), \qquad (8)$$

where $\gamma, \beta \in \mathbb{R}^C$ and $\lambda$ are instance-wise weights sampled from the Beta distribution. Subsequently, the style features are passed through a projector to get the final style condition weight.

## 4 EXPERIMENTS

### 4.1 Experimental Setup

**Datasets & Protocols.** Following the protocols established in prior works [29], we employ two distinct methods to evaluate the model's

generalization performance. In *Protocol 1*, we utilize four benchmark datasets: **I**diap-Replay-Attack (**I**)[3], **C**ASIA-FASD (**C**)[48], **M**SU-MFSD (**M**)[4], and **O**ULU-NPU (**O**)[1] for cross-dataset testing. In *Protocol 2*, we assess the model's effectiveness on the large-scale face anti-spoofing (FAS) datasets: CASIA-**S**URF (**S**)[46], CASIA-SURF **C**eFA (**C**)[14], and **W**MCA (**W**)[6]. Additionally, we report the experimental results obtained when using CelebA-Spoof [47] as a supplementary source dataset.

**Evaluation Metrics.** Two key metrics are utilized for assessing model performance: (1) Half Total Error Rate (HTER), which calculates the mean of the False Rejection Rate (FRR) and the False Acceptance Rate (FAR). (2) Area Under Curve (AUC), assessing the model's theoretical effectiveness.

**Implementation Details.** We pre-process all face images to a size of 224×224×3 and split them into patches of size 14×14. The image and text encoders are adapted from the pre-trained ViT-B/16 model of CLIP. For the text input, we set a text prompt template for each of the real and fake classes. Prompt token vectors and the adapter network are both initialized with $\mathcal{N}(0, 0.01)$. We have established 77 as the maximum number of textual tokens, with a vector dimension $d_t = 512$. The dimension of the image representations is $d_v = 768$. Our approach is implemented using PyTorch and trained using the Adam optimizer with a learning rate of $10^{-6}$ and a batch size of 32.

### 4.2 Comparisons to Prior Arts

To illustrate our model's ability to adapt to unseen domains, we employ Leave-One-Out (LOO) validation and give the Protocol 1 result summarizes in Tab. 1. This process involves executing cross-domain generalization within four typical LOO scenarios for the FAS task. In this framework, we randomly choose three datasets as source domains, and the fourth dataset is designated as the unseen target domain. Tab. 1 categorizes the comparison methods into two distinct groups: conventional DG FAS approaches [2, 8, 17, 18, 27, 28, 31, 34, 35, 37, 42] and CLIP-based techniques [26, 49, 50]. Notably, all outcomes are derived without employing the CelebA-Spoof [47] dataset as an additional resource. All results obtain without using CelebA-Spoof as the supplementary dataset.

From the Tab. 1, we have the following observations. (1) S-CPTL achieves the lowest average Half Total Error Rate (HTER) of 3.33% across all four cross-dataset testing scenarios, outperforming the second-best method, CoOp, by a significant margin of 1.04%. This highlights the robustness and generalization ability of your proposed approach. Furthermore, S-CPTL obtains the best HTER and AUC values in three out of four testing scenarios (OCI → M, OMI → C, and ICM → O), demonstrating its strong performance across different domain combinations. (2) It is worth noting that even in the OCM → I scenario, where S-CPTL does not achieve the best HTER, it still outperforms most of the compared methods and obtains the highest AUC of 98.63%.

In Table 2, we present a comparative analysis of our proposed S-CPTL method against several state-of-the-art baseline methods, including ViT [7], CLIP-V [26], CLIP [26], CoOp [50] and CoCoOp [49], on Protocol 2. CLIP-V represents a variant of CLIP with the text encoder removed. Our S-CPTL method demonstrates superior performance across all sub-protocols, significantly reducing the Half Total Error Rate (HTER) compared to CoOp for the target domains

**Table 1: Comparison with state-of-the-art FAS methods across Idiap-Replay-Attack (I), CASIA-FASD (C), MSU-MFSD (M) and OULU-NPU (O) datasets of Protocol 1. The bold numbers highlight the best performance.**

| Method | OCI → M | | OMI → C | | OCM → I | | ICM → O | | avg. |
|---|---|---|---|---|---|---|---|---|---|
| | HTER(%) | AUC(%) | HTER(%) | AUC(%) | HTER(%) | AUC(%) | HTER(%) | AUC(%) | HTER(%) |
| MADDG(CVPR'19) [27] | 17.69 | 88.06 | 24.50 | 84.51 | 22.19 | 84.99 | 27.98 | 80.02 | 23.09 |
| DR-MD-Net(TFIS'2020) [34] | 17.02 | 90.10 | 19.68 | 87.43 | 20.87 | 86.72 | 25.02 | 81.47 | 20.64 |
| RFMata(AAAI'20) [28] | 13.89 | 93.98 | 20.27 | 88.16 | 17.30 | 90.48 | 16.45 | 91.16 | 16.97 |
| NAS-FAS(TPAMI'20) [42] | 19.53 | 88.63 | 16.54 | 90.18 | 14.51 | 93.84 | 13.80 | 93.43 | 16.09 |
| D2AM(AAAI'21) [2] | 12.70 | 95.66 | 20.98 | 85.58 | 15.43 | 91.22 | 15.27 | 90.87 | 16.09 |
| SDA(AAAI'21) [35] | 15.40 | 91.80 | 24.50 | 84.40 | 15.60 | 90.10 | 23.10 | 84.30 | 19.65 |
| DRDG(IJCAI'21) [18] | 12.43 | 95.81 | 19.05 | 88.79 | 15.56 | 91.79 | 16.63 | 91.75 | 15.66 |
| ANRL(ACM MM'21) [17] | 10.83 | 96.75 | 17.83 | 89.26 | 16.03 | 91.04 | 15.67 | 91.90 | 15.09 |
| SSDG-R(CVPR'20) [8] | 7.38 | 97.17 | 10.44 | 95.94 | 11.71 | 96.59 | 15.61 | 91.54 | 11.28 |
| SSAN-R(CVPR'22) [37] | 6.67 | 98.75 | 10.00 | 96.67 | 8.88 | 96.79 | 13.72 | 93.63 | 9.81 |
| PatchNet(CVPR'22) [31] | 7.10 | 98.46 | 11.33 | 94.58 | 13.40 | 95.67 | 11.82 | 95.07 | 10.91 |
| SA-FAS(CVPR'23) [30] | 5.95 | 96.55 | 8.78 | 95.37 | 6.58 | 97.54 | 10.00 | 96.23 | 7.82 |
| IADG(CVPR'23) [52] | 5.41 | 98.19 | 8.70 | 96.44 | 10.62 | 94.50 | 8.86 | 97.14 | 8.39 |
| CLIP-V(PMLR'2021) [26] | 4.29 | 98.76 | 5.00 | 98.89 | 7.14 | 97.92 | 6.09 | 98.12 | 5.63 |
| CLIP(PMLR'2021) [26] | 4.04 | 99.13 | 5.00 | 98.89 | 6.57 | 98.45 | 6.09 | 98.12 | 5.43 |
| CoOp(IJCV'2022) [50] | 3.86 | 99.08 | 2.33 | 98.92 | **6.07** | 98.52 | 5.83 | 98.97 | 4.37 |
| CoCoOp(CVPR'2022) [49] | 4.16 | 99.01 | 5.17 | 98.19 | 6.21 | 98.50 | 6.00 | 98.49 | 5.39 |
| **S-CPTL(Ours)** | **1.43** | **99.17** | **0.89** | **99.00** | 6.86 | **98.63** | **4.12** | **99.02** | **3.33** |

**Table 2: Comparison with state-of-the-art FAS methods across results on CASIA-SURF (S), CASIA-SURF CeFA (C), and WMCA (W) datasets of Protocol 2. Note that the ∗ indicates the corresponding method using CelebA-Spoof as the supplementary source dataset. Bold numbers highlight the best performance.**

| Method | CS → W | | SW → C | | CW → S | | avg. |
|---|---|---|---|---|---|---|---|
| | HTER(%) | AUC(%) | HTER(%) | AUC(%) | HTER(%) | AUC(%) | HTER(%) |
| ViT [7] | 22.18 | 89.76 | 17.59 | 89.71 | 17.11 | 90.46 | 18.96 |
| CLIP-V [26] | 21.88 | 88.49 | 17.00 | 90.24 | 17.05 | 92.97 | 18.64 |
| CLIP [26] | 16.74 | 89.99 | 15.31 | 88.75 | 14.01 | 96.45 | 15.35 |
| CoOp [50] | 12.00 | 93.74 | 15.12 | 89.05 | 10.46 | 96.73 | 12.53 |
| CoCoOp [49] | 13.89 | 90.74 | 15.49 | 89.40 | 13.76 | 95.59 | 14.38 |
| **S-CPTL(Ours)** | **8.99** | **94.01** | **12.78** | **91.64** | **9.48** | **95.83** | **10.42** |
| ViT* [7] | 7.98 | 97.97 | 11.13 | 95.46 | 13.35 | 94.13 | 10.82 |
| FLIP-MCL* [29] | 4.46 | 99.16 | 9.66 | 96.69 | 11.71 | 95.21 | 8.61 |
| **S-CPTL*(Ours)** | **4.42** | **99.30** | **9.59** | **96.73** | **10.97** | **97.40** | **8.33** |

**Table 3: Ablation study of each component across four datasets on Protocol 1. The bold numbers highlight the best performance.**

| Components | | | OCI → M | | OMI → C | | OCM → I | | ICM → O | | avg. |
|---|---|---|---|---|---|---|---|---|---|---|---|
| PTL | AD | HSAM | HTER(%) | AUC(%) | HTER(%) | AUC(%) | HTER(%) | AUC(%) | HTER(%) | AUC(%) | HTER(%) |
| | | | 6.69 | 95.06 | 7.02 | 97.56 | 9.86 | 94.37 | 6.78 | 97.89 | 7.59 |
| ✓ | | | 2.62 | 98.57 | 3.43 | 97.88 | 8.57 | 97.00 | 4.72 | 98.01 | 4.84 |
| ✓ | ✓ | | 1.43 | 99.13 | 1.44 | 98.14 | 6.98 | 98.45 | 4.23 | 98.88 | 3.52 |
| | ✓ | | 4.35 | 98.54 | 3.46 | 97.99 | 7.08 | 97.99 | 5.77 | 99.01 | 5.21 |
| | ✓ | ✓ | 4.29 | 98.76 | 3.40 | 98.89 | 6.97 | 97.92 | 5.19 | 98.05 | 4.96 |
| ✓ | ✓ | ✓ | **1.43** | **99.17** | **0.89** | **99.00** | **6.86** | **98.63** | **4.12** | **99.02** | **3.33** |

**Table 4: Ablation study of depth of prompt token across four datasets on Protocol 1. The bold numbers highlight the best performance.**

| Deepth | OCI → M | | OMI → C | | OCM → I | | ICM → O | | avg. |
|--------|---------|---------|---------|---------|---------|---------|---------|---------|---------|
|        | HTER(%) | AUC(%)  | HTER(%) | AUC(%)  | HTER(%) | AUC(%)  | HTER(%) | AUC(%)  | HTER(%) |
| $J = 2$  | 1.43 | 99.41 | 2.11 | 99.43 | 10.36 | 96.09 | 3.25 | 98.44 | 4.29 |
| $J = 5$  | 1.43 | 99.46 | 1.44 | 99.25 | 8.93 | 97.1 | 4.2 | 98.89 | 4.00 |
| $J = 9$  | **1.43** | 99.17 | **0.89** | 99.00 | **6.86** | **98.63** | **4.12** | **99.02** | **3.33** |
| $J = 12$ | 1.43 | **99.73** | 1.22 | **99.54** | 9.71 | 96.43 | 4.32 | 99.15 | 4.17 |

CS→W, SW→C, and CW→S from 12.00%, 15.12%, and 10.46% to 8.99%, 12.78%, and 9.48%, respectively. S-CPTL achieves an average HTER of 10.42%, outperforming all baseline methods. Interestingly, CoCoOp exhibits suboptimal performance compared to CoOp, with average HTERs of 14.38% and 12.53%, respectively. While CoCoOp typically excels in recognition tasks with well-defined semantic categories, we hypothesize that in FAS tasks, the ambiguous semantic categories and lack of semantic information for CLIP lead to CoCoOp's inferior performance compared to CoOp. By incorporating the CelebA-Spoof [47] dataset into the training data, our S-CPTL* variant maintains its superior performance, achieving an HTER of 4.42% and an AUC of 99.30% for the target domain W, surpassing ViT*[7] and FLIP-MCL*[29]. The average HTER of S-CPTL* is 8.33%, which is lower than both ViT* (10.82%) and FLIP-MCL* (8.61%), further validating the effectiveness of our proposed approach.

## 4.3 Ablation Studies

**Effectiveness of each component.** We conducted several experiments to show the advantage of each component in our S-CTPL using the same protocol as above. The experimental results of various combinations are listed in Tab. 3. In the first row, no anything means that we just employ the CLIP to capture spoof feature for FAS, which we define as baseline. The baseline only MLP head is fine-tuned, the rest is frozen. It is a pity that the perform not well in unseen domains. The performance of the baseline with prompt token learning (PTL) is substantially better than the baseline in all cases, i.e., for in HTER, +4.07% (OCI → M), +3.59% (OMI → C), +1.92%(OCM → I),+1.98%(ICM → O). PTL plays an indispensable role in the application of CLIP to downstream FAS tasks because PTL focuses on capturing fine-grained spoof feature in shallow layers to enhance the feature representation. Adapter (AD) leverages the adapter scheme to to efficiently fine-tune a pre-trained ViT in CLIP for cross-domain generalized FAS. Adding an adapter to a baseleine with PTL increases performance by an average of 1.32% over a PTL-only structure. Hybrid style augmentation module (HSAM) is introduced to extract the style feature and weaken it's influence for domain generalization. Moreover, Mixed style increase their diversity through mixed feature statistics. HSAM with adapter achieves an average 0.25% improvement of four datasets. In summury, the goal of all component is to assist the base network in learning the complete and robust spoof feature. So the last row of our whole S-CTPL achieves the best results in various cases.

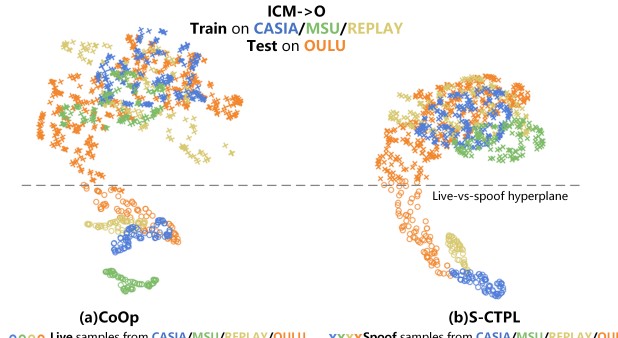

**Figure 4: visualization for the feature learned from the penultimate layer of the proposed S-CTPL method in the cross-dataset FAS task of** ICM → O

**Effectiveness of depth of prompt token.** To investigate the impact of hierarchical contextual representations, we embed learnable prompt tokens within the initial $J$ (where $J < K$) layers of the language branches. Tab. 4 presents an ablation study on the depth of prompt tokens across four datasets using Protocol 1. The results demonstrate that increasing the prompt token depth generally improves performance on base classes, but at the cost of reduced accuracy on novel classes. The best performance is achieved when $J = 9$, with an average HTER of 3.33% and the lowest individual HTERs of 1.43%, 0.89%, 6.86%, and 4.12% for the target domains OCI→M, OMI→C, OCM→I, and ICM→O, respectively. However, when $J = 12$, there is a notable increase in HTER, suggesting that excessive depth leads to overfitting and hinders the model's ability to generalize effectively to novel classes.

## 5 CONCLUSION

In this work, we propose S-CTPL, an efficient prompt token learning framework for adapting VLMs to generalizable FAS. We underline the importance of prompt token learning to achieve complex and label-non-semantic task (i.e. FAS). Through a style condition generation structure, we form multi-level style projection which carries instance-specific visual features as the condition. As a result, we align the visual-language representation to produce state-of-the-art performance on 0-shot FAS classification.

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
