# OpenReview forum: "Style-conditional Prompt Token Learning for Generalizable Face Anti-spoofing"
_acmmm.org/ACMMM/2024/Conference — MM2024 Poster_

### Official Review · Reviewer_CeTM · 2024-05-07

**Rating:** 5
**Confidence:** 4

**Summary:**

This paper proposed a novel approach called Style-Conditional Prompt Token Learning (S-CPTL) for domain generalization face anti-spoofing. Instead of directly manipulating visual features, S-CPTL utilizes large-scale vision-language models (VLMs) and leverages the textual feature to dynamically adjust the classifier’s weights for exploring generalizable visual features. This method utilizes training the introduced prompt tokens to carry visual styles and use them as weights for classifiers to improve the model's generalization Extensive experiments demonstrate the advantages of S-CTPL over state-of-the-art methods.

**Strengths:**

1. The S-CTPL proposes a novel style conditional prompt framework that adaptively capture live-irrelevant signals from the instance-specific styles and increase their diversity through mixed feature statistics to further reduce the overfitting of the model.
2. The learnable prompt token and adapter constitute a resource-efficient technique, as they operate with a limited set of parameters and reduced computational requirements.
3. The proposed method sets a new state-of-the-art benchmark.

**Limitations:**

1.The benefit about preventing the absence of semantic information and disruption of semantic structure in introduction, how to understand the sentence? Please give some explain “absence of semantic information” and “disruption of semantic structure”.
2.The description of the SCM in Section 3.4 of the article does not match the structure in Fig.2.
3.The motivation behind the hybrid style augmentation module (HSAM) requires detailed explanation, and this aspect also needs further analysis in the experiments.

**Suitability:**

3

---

### Official Review · Reviewer_vGUK · 2024-05-24

**Rating:** 3
**Confidence:** 3

**Summary:**

This paper tackles generalizable face anti-spoofing by using the proposed Style-Conditional Prompt Token Learning to learn dynamic prompt tokens that capture generalizable representations for improved generalizability.

**Strengths:**

- This paper introduces prompt token learning for FAS tasks, which reduces the semantic gap in the model by leveraging generalizable language information for visual classification.
- The method achieves satisfactory performance on several testing protocols, which verifies the effectiveness of the newly proposed Style-Conditional Prompt Token Learning framework.
- The ablation studies further demonstrate the contribution of each proposed component, which provides support for the overall performance gain.

**Limitations:**

- Table 1 of the manuscript does not compare some recent methods, such as FLIP [29], UDG-FAS [a], and DiVT-M [b].
- The Mixstyle technique used in the proposed Style Condition Module has been explored in many prior FAS works, including [17,37,52], which reduces the novelty of the proposed technique. Further, does the method leverage domain labels for the Style Condition Module in the experiments?
- To reflect the proposed method's true generalizability, the average results, along with the standard deviations, on multiple runs under different seeds are expected to be included (similar to FLIP [29]).
- Rather than simply presenting numeric results, a more in-depth analysis should be conducted to demonstrate the method's effectiveness. I suggest the authors present further analysis of the optimized prompt tokens, the method's attention maps, and some failure cases to help readers understand its performance intuitively.
- Some references do not show correctly in the manuscript. For instance, in L121-L122, " CoOp [?] and CoCoOp [?]". Please resolve this in the revision.
- Some texts are duplicated in the manuscript. For instance, in L138-143, "Furthermore, since liveness-irrelevant cues are lying within style features, the key to FAS domain generalization is to prevent the models from overfitting to instance-specific styles." appears twice. A similar issue can be found In L146-L152.
- What's the meaning of "label-non-semantic task" in L807 in Section 5?

[a] Towards Unsupervised Domain Generalization for Face Anti-Spoofing. ICCV 2023.

[b] Domain Invariant Vision Transformer Learning for Face Anti-Spoofing. WACV 2023.

**Suitability:**

3

---

### Official Review · Reviewer_Ygdk · 2024-05-26

**Rating:** 4
**Confidence:** 4

**Summary:**

This paper proposes a FAS method based on domain generalization named Style-Conditional Prompt Token Learning (S-CPTL). The method aims to improve the model's generalization ability by training the introduced prompt tokens to carry visual style and using them as the classifier's weights. Compared with static prompt tokens, S-CPTL introduces dynamic prompt tokens that can adaptively capture the liveness signals that are independent of the instance-specific style, and increase the diversity of these signals by mixing feature statistics, thereby further reducing the model's overfitting. The effectiveness of the proposed method has been verified in various settings, including zero-shot cross-domain generalization and detection of unseen attacks.

**Strengths:**

1. The article constructs a prompt token learning framework for the FAS task. It can reduce the semantic gap within the model by using the general language guidance from large-scale models as the classification weights of visual features, thereby reducing the semantic gap within the model.
2. Generate style-conditional for a specific instance. We extract learnable style information and adaptively capture signals that are irrelevant to the scene. Compared with the inherent static methods, S-CPTL can dynamically mix feature statistics with text representations, thereby
further reducing the model's overfitting.

**Limitations:**

1. The performance of the SCM module proposed in this paper is highly contingent upon the weights derived from the mixed feature statistics sampled from the training data. Could this be influenced by the imbalance or biases inherent in the training data, potentially confining the statistical significance to the mixed statistics within the training dataset, rather than possessing a broader statistical meaning? Does this restrict the module's generalization capability to a wider array of unknown and novel types of deception? Are there more in-depth research details and discussions on this subject?
2. Some spoofing attacks, such as funny glasses and tattoos, may have textual descriptions that conflict with their spoofing functions. Could this lead to some conflict between the tokens in  the textual branch and those extracted in the visual branch?
3. The paper explores the impact of the depth of prompt tokens on generalization, noting that increasing the depth of prompt tokens generally enhances performance on base classes but at the expense of reduced accuracy on novel classes. Is this phenomenon due to the statistics described in point 1 or is it a result of the model's own fitting capabilities? The deepening of depth leading to suppression of new classes, does this imply that the proposed SCM module is particularly sensitive to certain factors (such as statistics, parameters, feature extraction capabilities)? Are there any potential limitations on its generalization ability?
4. There is a typographical error inline 219; "domian" should be corrected to "domain".

**Suitability:**

3

---

### Official Review · Reviewer_Fy59 · 2024-05-26

**Rating:** 4
**Confidence:** 4

**Summary:**

This paper proposed a Style-Conditional Prompt Token Learning (S-CPTL) method for face anti-spoofing by training the introduced prompt tokens to carry visual styles. Experiments on several datasets show that the proposed model achieves state-of-the-art performance.

**Strengths:**

- The motivation of this paper is clear and the authors chose a straightforward but effective method to achieve the goal.
- The combination of the dynamic prompt token and multi-level style projection for face anti-spoofing is interesting.
- The charts related to the proposed method and experiments in the paper are relatively clear, and the organization of the charts is logical.

**Limitations:**

- It is worth noting that there is a paper [1] submitted to arXiv one month before the submission deadline of ACMM, which is very consistent with the paper on arxiv in terms of motivation, pipeline, experimental setup and other aspects. We hope the authors can provide a detailed discussion to explain the relationship and differences between the two papers.

[1] Liu, Ajian, Shuai Xue, Jianwen Gan, Jun Wan, Yanyan Liang, Jiankang Deng, Sergio Escalera, and Zhen Lei. "CFPL-FAS: Class Free Prompt Learning for Generalizable Face Anti-spoofing." arXiv preprint arXiv:2403.14333 (2024).
- More recently published related works such as [1] and [2] about face anti-spoofing should be cited and discussed.

[2] Liu, Yuchen, Yabo Chen, Mengran Gou, Chun-Ting Huang, Yaoming Wang, Wenrui Dai, and Hongkai Xiong. "Towards unsupervised domain generalization for face anti-spoofing." In Proceedings of the IEEE/CVF International Conference on Computer Vision, pp. 20654-20664. 2023.

[3] Guo, Xiao, Yaojie Liu, Anil Jain, and Xiaoming Liu. "Multi-domain learning for updating face anti-spoofing models." In European Conference on Computer Vision, pp. 230-249. Cham: Springer Nature Switzerland, 2022.

[4] Yu, Bingyao, Jiwen Lu, Xiu Li, and Jie Zhou. "Salience-aware face presentation attack detection via deep reinforcement learning." IEEE Transactions on Information Forensics and Security 17 (2021): 413-427.
- Please unify the format of references. At least ensure that the citation formats of conferences and journals are consistent.
- This paper is completely inadequate for the discussion of Table 1. Further, the authors did not give any explanations for worse performance on the “OCM → I” benchmark compared with CoOp(IJCV’2022) [50].

In summary, if the authors can provide good answers to these questions, I will consider changing the ranking.

**Suitability:**

3

---

### Meta-Review · Area_Chair_kYBJ · 2024-07-08

**Recommendation:** Accept (Poster)
**Confidence:** 4

**Metareview:**

Four of the reviewers have arrived at the agreement of weak acceptance of this paper. All the reviewers mark the strength of the paper with token prompting techniques used into FAS problem. Most of the concerns are addressed and acknowledged by the reviewers. Though there is one concern remaining: the authors claimed Face Anti-Spoofing is a non-semantic task while the reviewer believe FAS require semantic knowledge to perform generalizable classification, it would be good that the authors avoid such discussion from the final paper version, while only focusing on the factual performance report.